# Food Insecurity and Sleep-Related Problems in Adolescents: Findings from the EHDLA Study

**DOI:** 10.3390/nu16121960

**Published:** 2024-06-20

**Authors:** Edina Maria de Camargo, Héctor Gutiérrez-Espinoza, José Francisco López-Gil

**Affiliations:** 1Department of Physical Education, Universidade Federal do Paraná (UFPR), Curitiba 81531-980, Brazil; 2One Health Research Group, Universidad de Las Américas, Quito 170124, Ecuador; 3Department of Communication and Education, Universidad Loyola Andalucía, 41704 Seville, Spain

**Keywords:** sleep, sleep disorders, food security, poverty, youth, teenagers

## Abstract

Purpose: The current research aimed to investigate the connection between food insecurity and sleep issues among Spanish adolescents aged from 12 to 17 years from the *Valle de Ricote* (Region of Murcia, Spain). Methods: Data from the Eating Healthy and Daily Life Activities Study, which included a sample of 836 adolescents (55.3% girls), were analyzed. Food insecurity was evaluated using the Child Food Security Survey Module in Spanish (CFSSM-S), while sleep-related problems were evaluated using the Bedtime problems, Excessive daytime sleepiness, Awakenings during the night, Regularity and duration of sleep, and Sleep-disordered breathing (BEARS) sleep screening tool. Generalized linear models were employed to explore the association between food insecurity and sleep-related issues. Results: Compared with their counterparts with food security, adolescents with food insecurity had greater probabilities of bedtime problems (24.1%, 95% confidence interval (CI) 16.9% to 33.0%, *p* = 0.003), excessive daytime sleepiness (36.4%, 95% CI 27.5% to 46.3%, *p* < 0.001), awakenings during the night (16.7%, 95% CI 10.8% to 25.1%, *p* = 0.004), and any sleep-related problems (68.1%, 95% CI 57.5% to 77.1%, *p* < 0.001). Conclusions: This study suggests that food insecurity is related to greater sleep-related problems among adolescents. Implementing strategies to mitigate food insecurity may contribute to improved sleep health among adolescents, highlighting the importance of integrated public health interventions.

## 1. Introduction

Food insecurity is a complex issue that can arise from a scarcity of food, both in terms of quantity and quality. Insufficient food and inadequate food diversity are key factors that contribute to food insecurity [1]. In addition, there are several other reasons for food insecurity. For instance, food may be physically unavailable in a particular region or country, making it difficult for individuals to obtain it. Moreover, food may be unaffordable, even if it is accessible on the market. Finally, an unequal distribution of food within households can also contribute to food insecurity. It is important to address these underlying factors to effectively combat food insecurity [2]. The consequences of food insecurity are a worsening quality of life, hunger, and the emergence of diseases and other physical and mental health problems [3]. All of these factors can lead to anemia, malnutrition [4], overweight (due to the consumption of foods that are poor in nutrients and rich in fat and sugar) [5], and can impair the immune system (e.g., reduced exocrine secretion of protective substances, impaired intestinal barrier function, and low plasma complement levels [6]). Food insecurity is a global problem, and the rate is estimated to be 27% [7], which is even greater among adolescents than among individuals in other subpopulations (moderate 44.9% and severe 6.2%) [8].

On the other hand, sleep quality and patterns can be significantly impacted by food and nutrient intake [9]. Sleep not only serves a restorative function, but also plays a central role in regulating metabolism, influencing appetite control, and affecting immunological functions [9]. Consuming a high-protein meal or excessive amounts of fatty foods before bedtime or late-night snacking on sugary foods can lead to poor sleep quality [10]. On the other hand, a well-balanced diet that includes a variety of nutrients, such as those found in fruits and vegetables, can improve sleep quality and overall well-being [11]. Research on the relationship between food insecurity and sleep-related problems has increased in the literature [12]. For instance, in a study by Wang et al. [12] that included 223,561 adolescents from 68 countries, the prevalence of severe food insecurity and sleep disturbance was 6.4% and 8.0%, respectively. Severe food insecurity was significantly associated with a greater risk of sleep disturbance in 48 countries. However, the measurement of food insecurity was based on a single question asking about the frequency of hunger caused by a lack of food consumed at home. Thus, this classification of food insecurity used may not accurately reflect its relationship with sleep disturbance [12]. The relationship between food insecurity and sleep-related problems in adolescents may be a public health issue, as both are associated with adverse health outcomes. Therefore, further research is necessary.

The impact of food insecurity is worldwide, so it is necessary that research into sleep-related problems be conducted with samples from countries that have not yet been studied to clarify whether this relationship extends to other adolescent samples. In Spain, food insecurity has increased as a consequence of the economic crisis and precarious employment [13]. In this context, previous studies have highlighted a high prevalence of food insecurity (using the Child Food Security Survey Module in Spanish [CFSSM-S]), ranging from 18.3% [14] to 19.2% [15]. In addition, another study (using the Household Food Insecurity Access Scale [HFIAS]) indicated a prevalence of 7.7% in adolescents. Based on this worrisome prevalence, it has been recommended to revise and strengthen public policies in Spain to provide more effective protection for children and families experiencing severe poverty to prevent long-term harm [16].

Along with irregular and insufficient sleep, food insecurity can disrupt daily activity rhythms, increase fatigue, hinder physical activity engagement, and foster a sedentary lifestyle, which can lead to a greater risk of disease [5]. The relationship between food insecurity and sleep-related problems, including lifestyle factors such as physical activity level and sedentary behavior, may provide valuable contributions to scientific knowledge in this area [5]. Therefore, to address the potential impact of food insecurity on sleep and its potential health consequences, this study assessed the association between food insecurity and sleep-related problems in a sample of Spanish adolescents. By focusing on this population, this study aimed to provide insights that could inform targeted interventions and policies. Ultimately, understanding these relationships could help to mitigate the adverse effects of food insecurity on adolescent health and well-being, emphasizing the importance of comprehensive strategies that address both nutritional and sleep health in vulnerable populations.

## 2. Materials and Methods

### 2.1. Study Design and Population

This study performed a secondary analysis using data from the Eating Healthy and Daily Life Activities (EHDLA) study, which included adolescents from the *Valle de Ricote* (Region of Murcia, Spain). Data were gathered during the 2021–2022 academic year from all three secondary schools in that area. The methodology of the EHDLA study was previously published [17]. Of the initial 1378 adolescents (100.0%) from the EDHLA study, 460 (33.4%) were removed from the study due to missing information about food insecurity. Additionally, 36 participants (2.6%) were eliminated due to a lack of data regarding sleep-related problems. Furthermore, 46 participants (3.3%) were excluded because of missing information on any covariate (i.e., body mass index, sedentary behavior, or energy intake). Therefore, the current analysis included 836 adolescents, 55.3% of whom were girls.

### 2.2. Procedures

#### 2.2.1. Food Insecurity

Household food insecurity was assessed using the CFSSM-S [18]. This survey measures participants’ perceptions of food insecurity, covering concerns such as running out of food, relying on inexpensive food, an inability to eat a balanced diet, reducing portion sizes, eating less, skipping meals, going hungry, and going without food for an entire day. The CFSSM-S consists of nine questions evaluated on a 3-point Likert scale. Negative responses (“never”) are scored as zero points, while affirmative responses indicating moderate or high food insecurity (“sometimes” or “a lot”) are scored as one point. Food security levels were classified based on criteria from original research [19] and the US Department of Agriculture [20]. Households were classified as “food secure” (0–1 point), “low food security” (2–5 points), or “very low food security” (6–9 points). For further analyses, we collapsed this variable into “food security” (“food secure”) and “food insecurity” (“very low food security” and “low food security”).

#### 2.2.2. Sleep-Related Problems

The Bedtime problems, Excessive daytime sleepiness, Awakenings during the night, Regularity and duration of sleep, and Sleep-disordered breathing (BEARS) sleep screening tool [21] was employed to identify common sleep issues in preschoolers, children, and adolescents. Parents or legal guardians reported participants’ sleep-related problems. This screening tool aids in the early detection of sleep-related issues due to its ease of application and demonstrated adequate performance [22].

#### 2.2.3. Covariates

The age and sex of the adolescents were self-reported. Socioeconomic status was assessed with the Family Affluence Scale (FAS-III) [23], which is computed from six items: family car ownership, adolescents’ own bedroom, number of household computers, number of bathrooms, dishwasher ownership, and number of holidays taken outside of Spain in the past year. The FAS-III score ranges from 0 to 13 points. Sedentary behavior and physical activity data were collected using the Spanish-Youth Activity Profile (YAP-S) [24], with scores calculated by summing the scores of each section. Energy intake was assessed through a self-reported dietary habits survey previously validated for the Spanish population [25]. Adolescents’ body weight and height were measured accurately using an electronic scale (Tanita BC-545, Tokyo, Japan) and a portable height rod (Leicester Tanita HR 001, Tokyo, Japan). Body mass index (BMI) was calculated by dividing body weight in kilograms by the square of height in meters.

### 2.3. Statistical Analysis

Visual techniques (i.e., density and quantile–quantile plots) and statistical procedures (i.e., Shapiro–Wilk test) were used to assess the normal distribution of variables. This study reported medians and interquartile ranges (IQRs) for quantitative variables and frequencies (*n*) and percentages (%) for qualitative variables. Generalized linear models (GLMs) were used to verify the association of food insecurity with sleep-related problems and were fitted using robust methods (i.e., the “*Mqle*” method) to handle heteroscedasticity and outliers. The predicted probabilities (%) and 95% confidence intervals (CIs) for each sleep-related problem, based on food security status, were calculated. In addition, further GLMs were constructed to test the association between each specific item of the CFSSM-S using backward stepwise regression (in addition to the previously mentioned robust methods). Covariates included sex, age, socioeconomic status, BMI, sedentary behavior, physical activity, and energy intake. Statistical analyses were performed using the R statistical software (version 4.3.2) and RStudio (version 2023.12.1+402), with significance set at *p* < 0.05.

## 3. Results

Table 1 shows the characteristics of the study participants based on their food security data in relation to sociodemographic, lifestyle, anthropometric, and sleep-related problems. Of the total sample of 836 individuals, 134 (16.0%) had food insecurity and 702 (84.0%) had food security. Participants with food insecurity experienced more sleep-related problems (bedtime problems: 37.3%; awakenings during the night: 26.1%; regularity and duration of sleep: 38.8%; snoring: 8.2%; and any sleep-related problems: 78.4%) than their counterparts with food security did.

Figure 1 indicates the predicted probabilities of having each sleep-related problem (i.e., bedtime problems, excessive daytime sleepiness, awakenings during the night, regularity and duration of sleep, snoring, or any sleep-related problem) per additional point in the CFSSM–S after adjusting for several covariates (i.e., sex, age, socioeconomic status, BMI, sedentary behavior, physical activity, and energy intake). Furthermore, the ORs and 95% CIs of the GLMs are shown in Table 2. For each additional point in the CFSSM–S, higher probabilities were observed for bedtime problems (2.9%, 95% CI 1.2% to 4.6%, *p* < 0.001), excessive daytime sleepiness (3.2%, 95% CI 1.2% to 5.2%, *p* = 0.001), awakenings during the night (2.6%, 95% CI 1.3% to 3.9%, *p* < 0.001), regularity and duration of sleep (2.1%, 95% CI 0.1% to 4.0%, *p* = 0.037), and any sleep-related problems (5.8%, 95% CI 3.0% to 8.7%, *p* < 0.001). Although a greater probability of snoring was also identified for each additional point in the CFSSM-S (0.5%, 95% CI −0.5% to 1.5%), this difference was not significant (*p* = 0.346).

Figure 2 indicates the predicted probabilities of having each sleep-related problem (i.e., bedtime problems, excessive daytime sleepiness, awakenings during the night, regularity and duration of sleep, snoring, or any sleep-related problem) based on food insecurity status after adjusting for several covariates. In addition, the ORs and 95% CIs of the GLMs are displayed in Table 3. Compared with their counterparts with food security, adolescents with food insecurity had greater probabilities of bedtime problems (24.1%, 95% CI 16.9% to 33.0%), excessive daytime sleepiness (36.4%, 95% CI 27.5% to 46.3%), awakenings during the night (16.7%, 95% CI 10.8% to 25.1%), regularity and duration of sleep (26.2%, 95% CI 18.9% to 35.1%), snoring (10.1%, 95% CI 4.9% to 20.0%), and any sleep-related problems (68.1%, 95% CI 57.5% to 77.1%). However, only significant differences were identified for bedtime problems (*p* = 0.003), excessive daytime sleepiness (*p* < 0.001), awakenings during the night (*p* = 0.004), and any sleep-related problems (*p* < 0.001).

The GLMs (using backward stepwise regression) examining the association between each specific item of the CFSSM-S and having each sleep-related problem (adjusted for several covariates) are shown in Table 4. Feeling worried about running out of food and skipping meals were associated with sleep problems (*p* = 0.027 and *p* = 0.036, respectively). Feeling worried (about running out of food) and being hungry were related to excessive daytime sleepiness (*p* = 0.017 and *p* = 0.013, respectively). A lack of balanced meals was linked to awakenings during the night (*p* < 0.001). Eating smaller portions was related to regularity and duration of sleep (*p* = 0.031). Lastly, a lack of balanced meals and feeling hungry were associated with having any sleep-related problems.

## 4. Discussion

The results of this study generally suggest that adolescents who reported food insecurity experienced greater problems related to sleep, including problems at bedtime, awakening during the night, changes in regular sleep, sleep duration, snoring, and other sleep-related problems, than adolescents who reported food security. Furthermore, as food insecurity increased, the likelihood of sleep problems also increased. More specifically, feeling worried (about running out of food), being hungry, skipping meals, lacking balanced meals, and eating smaller portions were associated with sleep-related problems. The results of the present study are in accordance with the literature [26,27,28,29]. For instance, King [27] emphasized that eating patterns and sleep-related problems (problems sleeping, waking up at night, and daytime drowsiness) are related. Na et al. [28] revealed that food insecurity was associated with a compromised sleep quality; however, there was no association with sleep duration. On the other hand, De Jong et al. [29] investigated the determinants of a short sleep duration, warning that eating habits, regardless of age and sex, contribute to problems related to sleep duration. In contrast, in a longitudinal study by Lee et al. [26], female adolescents who lived with food insecurity were more likely to develop sleep-related problems than male adolescents were [26]. In a global study with 189,619 adolescents by Osei Bonsu et al. [30], the authors concluded that female adolescents were at a greater risk of moderate and severe sleep disorders in the context of food insecurity [30]. The absence of a consistent eating routine was associated with a shorter sleep duration in the study by Jong et al. [29]. According to the American Academy of Sleep Medicine (AAP), the daily sleep duration for teenagers ranges from 8 to 10 h [31].

The mechanisms by which food insecurity can cause sleep-related problems have been investigated in the literature [5,32]. Eating habits can significantly impact sleep quality, as they lead to hormonal changes related to sleep [32]. Consuming a diet high in processed and sugary foods can increase insulin and leptin levels, disrupting both behavioral and circadian rhythms and causing irregular sleep–wake patterns [33]. Additionally, consuming large amounts of carbohydrates can stimulate ghrelin, a hormone that can impair sleep quality and increase appetite [34]. The intake of certain nutrients can also have an impact on sleep quality by controlling the production of serotonin and melatonin [35]. Tryptophan is an essential amino acid that serves as a precursor for both serotonin and melatonin, which are connected to sleep and alertness [35]. As a result, consuming foods that limit tryptophan availability and interfere with the synthesis of serotonin and melatonin can negatively affect sleep regulation [35]. Moreover, the quality of macronutrients, in addition to their quantity, may contribute to the development of sleep disorders [36].

Other possible explanation for the findings of this study could be associated with the family environment. In homes experiencing food insecurity, adolescents’ health can be adversely impacted, which may subsequently impair their sleep [2]. For example, a low household income and unstable employment are significant predictors of food insecurity [37]. Food insecurity often results from socioeconomic challenges, such as financial instability or poverty, limiting access to food [38]. Additionally, mothers’ food-related behaviors and parenting play significant roles in adolescent food insecurity [39]. Parents dealing with food insecurity often face high levels of stress and anxiety, which can influence their parenting style and interactions with their children [40]. Also, it has been noted that adolescents from food-insecure households are less likely to experience “family assets”, including positive parenting, supportive parent interactions, and a safe home environment [41]. On the other hand, parents’ understanding of nutrition can impact the quality of food choices made within a food-insecure setting [42], and a higher diet quality may be associated with fewer sleep-related problems [43] (also noted in adolescents [44,45]).

Adolescents experiencing food insecurity often live in stressful environments [46,47], which could influence their mental health. This could be especially relevant for the adolescent population, as the association between household food insecurity and mental health is stronger for them than for younger children [48]. Worrying about where their next meal will come from or the overall financial situation of their family can lead to heightened stress and anxiety, which can disrupt sleep patterns [49]. Furthermore, the correlation between mental health problems and sleep quality among adolescents has been noted in the scientific literature [50]. In regard to this, a previous study revealed that food insecurity in adolescents was related to poorer mental health across 95 countries [46]. Supporting this idea, Paquin et al. [47] observed in their cohort study that a persistent high risk of food insecurity in childhood was associated with psychosocial problems later in adolescence after adjusting for confounders (e.g., low income). On the other hand, the cognitive and emotional impacts of food insecurity could also explain these findings. Food insecurity can impair cognitive functions [51,52,53,54] and emotional well-being [51,55], leading to difficulties in focusing, irritability, and mood swings. A systematic review by Royer et al. [54] revealed a correlation between greater food insecurity and poorer cognitive function across the life course. Similarly, a systematic review by Gallegos et al. [52] pointed out the strongest association between food insecurity and externalizing behaviors in the young population. Moreover, in Spain, Shankar-Krishnan et al. [14] found that adolescents with food insecurity exhibited a poorer psychological well-being, greater body dissatisfaction, and a greater drive for thinness. These cognitive and emotional disturbances could contribute to insomnia and other sleep-related issues.

This study has several limitations that need to be acknowledged. First, due to its cross-sectional design, it was not possible to establish causality; therefore, future longitudinal observational studies are necessary to determine whether food insecurity leads to sleep-related issues among adolescents. Second, the use of self-reported measures could lead to bias due to possible inaccuracies and overreporting by adolescents. Third, even though the analyses were adjusted for various covariates (such as sociodemographic, lifestyle, and anthropometric factors), residual confounding factors cannot be ruled out. However, the study also has several strengths, including the use of a large sample of Spanish adolescents, which is a relatively understudied group. However, the data pertain to a very specific region of Spain, so they are not generalizable to the rest of the Spanish population. Additionally, adjusting for multiple potential confounders enhances the robustness of the findings and provides a more accurate depiction of the association between food insecurity and sleep-related issues.

## 5. Conclusions

This study suggests that food insecurity is related to greater sleep-related problems among adolescents. Specifically, adolescents with greater food insecurity were more likely to have bedtime problems, excessive daytime sleepiness, awakenings during the night, or any sleep-related problems. The findings remained constant despite adjusting for anthropometric, lifestyle, or sociodemographic variables, which underscores the significant impact that food insecurity could have on sleep-related issues among adolescents. Implementing strategies to mitigate food insecurity may contribute to improved sleep health among adolescents, highlighting the importance of integrated public health interventions.

## Figures and Tables

**Figure 1 nutrients-16-01960-f001:**
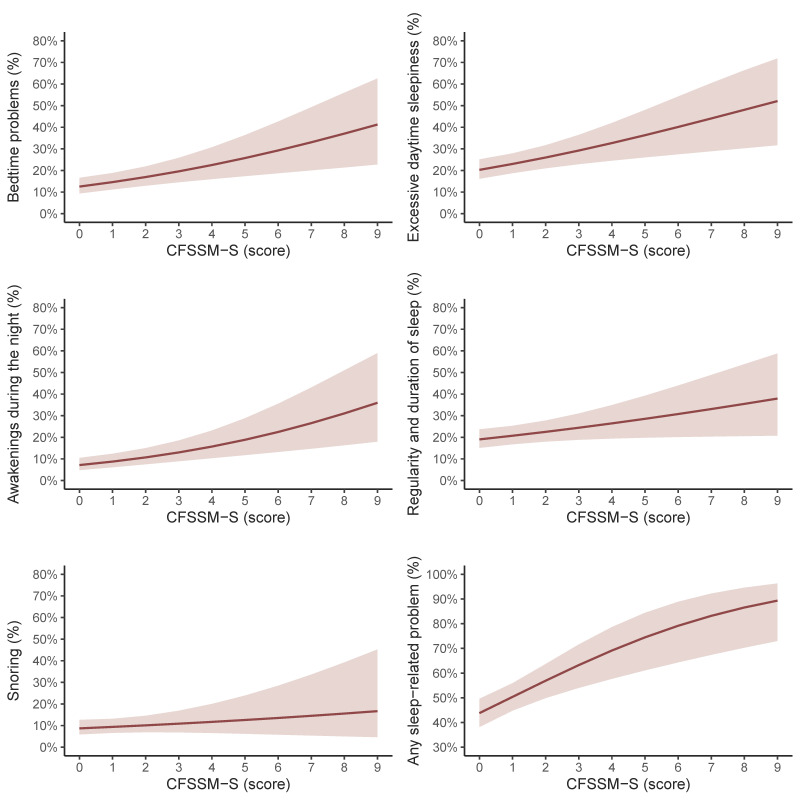
Predictive probabilities of having different sleep-related problems according to the Child Food Security Survey Module in Spanish (CFSSM-S) in adolescents. Adjusted for sex, age, socioeconomic status, body mass index, sedentary behavior, physical activity, and energy intake.

**Figure 2 nutrients-16-01960-f002:**
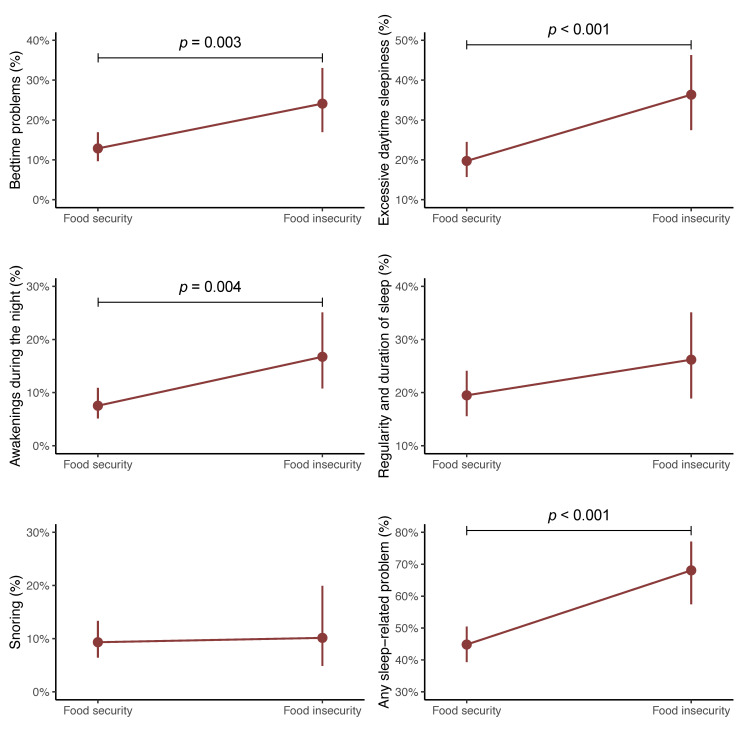
Predictive probabilities of having different sleep-related problems according to food security status in adolescents. Adjusted for sex, age, socioeconomic status, body mass index, sedentary behavior, physical activity, and energy intake.

**Table 1 nutrients-16-01960-t001:** Descriptive data of the study participants according to food insecurity status (*N* = 836).

Variables		Total Sample	Food Security	Food Insecurity	*p*-Value
Participants	*n* (%)	836 (100.0)	702 (84.0)	134 (16.0)	
Sex	Boys (%)	374 (44.7)	310 (44.2)	64 (47.8)	0.501
	Girls (%)	462 (55.3)	392 (55.8)	70 (52.2)	
Age (years)	Median (IQR)	14.0 (2.0)	14.0 (2.0)	14.0 (2.0)	0.897
FAS-III (score)	Median (IQR)	8.0 (3.0)	9.0 (3.0)	7.0 (3.0)	<0.001
YAP-S sedentary behaviors (score)	Median (IQR)	2.6 (0.8)	2.4 (0.8)	2.6 (0.8)	0.201
YAP-S physical activity (score)	Median (IQR)	2.6 (0.8)	2.6 (0.8)	2.7 (1.0)	0.003
Body mass index (kg/m^2^)	Median (IQR)	21.6 (6.0)	21.6 (5.9)	21.9 (6.9)	0.513
Energy intake (kcal)	Median (IQR)	2587.5 (1489.5)	2522.1 (1398.7)	3159.7 (2389.8)	<0.001
Bedtime problems	No (%)	643 (76.9)	559 (79.6)	84 (62.7)	<0.001
	Yes (%)	193 (23.1)	143 (20.4)	50 (37.3)	
Excessive daytime sleepiness	No (%)	560 (67.0)	494 (70.4)	66 (49.3)	<0.001
	Yes (%)	276 (33.0)	208 (29.6)	68 (50.7)	
Awakenings during the night	No (%)	711 (85.0)	612 (87.2)	99 (73.9)	<0.001
	Yes (%)	125 (15.0)	90 (12.8)	35 (26.1)	
Regularity and duration of sleep	No (%)	592 (70.8)	510 (72.6)	82 (61.2)	0.010
	Yes (%)	244 (29.2)	192 (27.4)	52 (38.8)	
Snoring	No (%)	780 (93.3)	657 (93.6)	123 (91.8)	0.566
	Yes (%)	56 (6.7)	45 (6.4)	11 (8.2)	
Sleep-related problems (number)	Median (IQR)	1.0 (2.0)	1.0 (2.0)	2.0 (1.0)	<0.001
Any sleep-related problem	No (%)	354 (42.3)	325 (46.3)	29 (21.6)	<0.001
	Yes (%)	482 (57.7)	377 (53.7)	105 (78.4)	

FAS-III; Family Affluence Scale-III; IQR, interquartile range; and YAP-S, Spanish Youth Active Profile.

**Table 2 nutrients-16-01960-t002:** Association between Child Food Security Survey Module in Spanish and sleep-related problems in adolescents.

Outcome		
Bedtime problems (yes)		
OR	95% CI	*p*-value
1.19	1.09 to 1.30	0.001
Excessive daytime sleepiness (yes)		
1.17	1.07 to 1.28	0.002
Awakenings during the night (yes)		
1.25	1.13 to 1.36	<0.001
Regularity and duration of sleep (yes)		
1.11	1.01 to 1.22	0.039
Snoring (yes)		
1.09	0.91 to 1.27	0.346
Any sleep-related problem (yes)		
1.30	1.17 to 1.43	<0.001

The odds ratios correspond to a one-point increase in the Spanish score of the Child Food Security Survey Module. Adjusted for age, sex, socioeconomic status, body mass index, physical activity, sedentary behavior, and energy intake. CI, confidence interval; and OR, odds ratio.

**Table 3 nutrients-16-01960-t003:** Association between food insecurity status and sleep-related problems in adolescents.

	Outcome		
	Bedtime problems (yes)		
Predictor	OR	95% CI	*p*-value
Food security	Reference		
Food insecurity	2.15	1.19 to 3.30	0.003
	Excessive daytime sleepiness (yes)		
Food security	Reference		
Food insecurity	2.32	1.25 to 3.50	<0.001
	Awakenings during the night (yes)		
Food security	Reference		
Food insecurity	2.50	1.39 to 4.01	0.004
	Regularity and duration of sleep (yes)		
Food security	Reference		
Food insecurity	1.47	0.97 to 2.21	0.068
	Snoring (yes)		
Food security	Reference		
Food insecurity	1.10	0.47 to 2.54	0.828
	Any sleep-related problem (yes)		
Food security	Reference		
Food insecurity	2.62	1.65 to 4.18	<0.001

Adjusted for sex, age, socioeconomic status, body mass index, sedentary behavior, physical activity, and energy intake. CI, confidence interval; and OR, odds ratio.

**Table 4 nutrients-16-01960-t004:** Generalized linear models with binomial distributions were constructed using backward stepwise regression to test the associations between each specific item of the Child Food Security Survey Module in Spanish and sleep-related problems in adolescents.

	Outcome					
	Bedtime Problems	Excessive Daytime Sleepiness	Awakenings During the Night	Regularity and Duration of Sleep	Snoring	Any Sleep-Related Problem
Predictor	OR (95% CI)	OR (95% CI)	OR (95% CI)	OR (95% CI)	OR (95% CI)	OR (95% CI)
Item 1: Worry	1.58 (1.05 to 2.36)	1.57 (1.08 to 2.28)	Removed at step 4	Removed at step 8	Removed at step 3	Removed at step 7
Item 2: Food run out	Removed at step 2	Removed at step 6	Removed at step 1	Removed at step 2	Removed at step 5	Removed at step 6
Item 3: Cheap food	Removed at step 7	Removed at step 2	Removed at step 5	Removed at step 5	Removed at step 6	Removed at step 2
Item 4: Balanced meal	Removed at step 5	Removed at step 3	3.56 (2.11 to 6.02)	Removed at step 1	Removed at step 2	1.87 (1.07 to 2.66)
Item 5: Eat less	Removed at step 1	Removed at step 1	Removed at step 2	2.03 (1.07 to 3.88)	Removed at step 4	Removed at step 4
Item 6: Meals cut	Removed at step 6	Removed at step 5	Removed at step 8	Removed at step 4	Removed at step 9	Removed at step 3
Item 7: Skip meal	2.34 (1.06 to 5.18)	Removed at step 6	Removed at step 7	Removed at step 3	Removed at step 8	Removed at step 5
Item 8: Hungry	Removed at step 4	2.34 (1.20 to 4.57)	Removed at step 3	Removed at step 7	Removed at step 7	3.19 (1.26 to 8.09)
Item 9: Not eat for a whole day	Removed at step 3	Removed at step 4	Removed at step 6	Removed at step 6	Removed at step 1	Removed at step 1

Variables with the highest *p*-value (above 0.05) were eliminated at each indicated step. Adjusted for sex, age, socioeconomic status, body mass index, sedentary behavior, physical activity, and energy intake. CI, confidence interval; and OR, odds ratio.

## Data Availability

The data used in this study are available upon request from the corresponding authors. However, given that the participants are minors, privacy and confidentiality must be respected.

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
