# Peer review of "Food Insecurity and Sleep-Related Problems in Adolescents: Findings from the EHDLA Study"

_nutrients, 2024, doi:10.3390/nu16121960_

Round 1

Reviewer 1 Report

Comments and Suggestions for Authors

The study evaluated the association between food insecurity and sleep related problems among 836 adolescents aged 12-17, living in a Spain region. Food insecurity resulted associated with the odd of sleep perturbation, if compared with food security.

Some small amendments:

Line 45: reduction in the immune system (functions? activity? efficiency?)

Line 106: after “food secure” the comma is missing.

Lines 232-233: the sentence “Additionally, consuming high amounts of carbohydrates can stimulate ghrelin, a hormone that plays a role in both sleep quality and appetite [30]” should be better declined. 

Line 258. “which could influence on their mental health”. Erase “on”

Lines 264-266: the sentence “In this sense, a previous study identified that adolescents with food insecurity was related to poorer mental health across 95 countries [42]” needs to be corrected. 

Lines 276-278: the sentence “found that poor psychological wellbeing, greater body dissatisfaction and higher drive for thinness in adolescents with food insecurity” needs to be corrected. 

I suggest improving the quality of the Figures and an English check.

Thank you for the opportunity to review this article. It is interesting and it is worthy for publication in terms of scientific and technical merit. It can be suitable for publication after minor revisions.

Author Response

REVIEWER 1

The study evaluated the association between food insecurity and sleep related problems among 836 adolescents aged 12-17, living in a Spain region. Food insecurity resulted associated with the odd of sleep perturbation, if compared with food security.

Some small amendments:

Line 45: reduction in the immune system (functions? activity? efficiency?)

It has been modified as follows: “All of this can lead to anemia, malnutrition [4], overweight (due to the consumption of foods which are poor in nutrients and rich in fat and sugar) [5], and can impair the immune system (e.g., reduced exocrine secretion of protective substances, impaired intestinal barrier function, and low plasma complement levels).”

Line 106: after “food secure” the comma is missing.

Done. Thank you.

Lines 232-233: the sentence “Additionally, consuming high amounts of carbohydrates can stimulate ghrelin, a hormone that plays a role in both sleep quality and appetite [30]” should be better declined. 

It has been modified as follows: “Additionally, consuming large amounts of carbohydrates can stimulate ghrelin, a hormone that can impair sleep quality and increase appetite”. Thank you.

Line 258. “which could influence on their mental health”. Erase “on”

Done. Thank you.

Lines 264-266: the sentence “In this sense, a previous study identified that adolescents with food insecurity was related to poorer mental health across 95 countries [42]” needs to be corrected. 

It has been modified as follows: “In this sense, a previous study identified that adolescents with food insecurity were related to poorer mental health across 95 countries”. Thank you.

Lines 276-278: the sentence “found that poor psychological wellbeing, greater body dissatisfaction and higher drive for thinness in adolescents with food insecurity” needs to be corrected. 

It has been modified as follows: “The study found that adolescents with food insecurity exhibited poorer psychological wellbeing, greater body dissatisfaction, and a higher drive for thinness”. Thank you.

I suggest improving the quality of the Figures and an English check.

 Done. Thank you.

Thank you for the opportunity to review this article. It is interesting and it is worthy for publication in terms of scientific and technical merit. It can be suitable for publication after minor revisions.

Thank you so much for your time and feedback.

Reviewer 2 Report

Comments and Suggestions for Authors

The issue of the relationship between food insecurity and sleep-related problems in adolescents is of scientific interest. After reading the manuscript I have several comments addressed to the authors:

Introduction

The authors report that the previous studies confirm the relationship between insecurity and sleep problems in adolescents, emphasizing the limitations of the method of measuring insecurity used in these studies. At the same time, they justify the need to conduct research in populations not yet recognized in this regard. However, there is a lack of information on what is known about this topic about the Spanish population. This makes the justification for the study presented in the Introduction not sufficiently convincing.

Materials and Methods

Study Design and Population

There is a cited work that describes the sample selection, although it is worth emphasizing that this article only includes the first phase of the study (without follow-up). The size of the study group also differs (1138 participants vs 836 participants). Please complete the text and provide appropriate explanations.

Procedures

Lines 105-107. In my opinion, creating a category of “food insecurity” (category “very low food security”, and “low food security” together) is not a good solution from the perspective of further analyses because it may make it difficult to see the differences between categories: food security vs food insecurity. It would be better to look for differences in sleep problems in the group of 1/ food security vs. very low food security and 2/ food security vs. low food security. It seems that further analyses could be more interesting if the variable food security was treated as a continuous variable.

Results

Lines 145-148. Are these differences significant? Why is there no p-value presented in the table, which would allow for an easy assessment of the obtained results?

Line 158. It is necessary to specify what these covariates were.

Line 158. A new paragraph should start here or Table 2 should be cited at the beginning because this information is presented there, and not in Figure 1.

Figure 1. The quality of this Figure should be improved, e.g. it is difficult to read the variable name descriptions.

Table 2. It may be worth rethinking the table layout. Is it necessary to repeat CFSSM-S (per one point) for each variable? I suggest adding CFSSM-S score to the CFSSM-S description.

Figure 1 and Table 2. Please explain whether these results are not about the same thing. Is it necessary to show them in two different graphic forms? Wouldn't a table be sufficient? A similar comment applies to Figure 2 and Table 3. Why is statistical significance not shown in Figure 1 as in Figure 2?

Discussion

The discussion is largely in the form of a literature review, e.g. lines 257-279, because there is no direct connection between this information and the results of our research. The difficulty in discussing the results may be related to using a general indicator to assess food security. It may be worth presenting the components of the CFSSM-S in more detail in the Result section, and then comparisons with the results of other studies will be easier.

Lines 287-288. The large group is a strength, but the selection of the group (small region) does not allow for generalization to the entire population of young Spaniards and this is a limitation of the study.

Conclusions

I suggest formulating conclusions in more detail, i.e. to indicate what sleep problems are associated with food insecurity.

Author Response

REVIEWER 2

The issue of the relationship between food insecurity and sleep-related problems in adolescents is of scientific interest. After reading the manuscript I have several comments addressed to the authors:

Thank you so much for your time and feedback.

Introduction

The authors report that the previous studies confirm the relationship between insecurity and sleep problems in adolescents, emphasizing the limitations of the method of measuring insecurity used in these studies. At the same time, they justify the need to conduct research in populations not yet recognized in this regard. However, there is a lack of information on what is known about this topic about the Spanish population. This makes the justification for the study presented in the Introduction not sufficiently convincing.

Regarding Spain, food insecurity has increased as a consequence of the economic crisis and precarious employment (Gracia-Arnaiz, 2022). In this context, previous studies have highlighted a high prevalence of food insecurity (using the Child Food Security Survey Module in Spanish (CFSSM-S)), ranging from 18.3% (Shankar-Krishnan et al., 2021) to 19.2% (Barreiro-Álvarez et al., 2024). In addition, another study (using Household Food Insecurity Access Scale (HFIAS)) indicated a prevalence of 7.7% in adolescents. Based on this worrisome prevalence, it has been recommended to revise and strengthen public policies in Spain to provide more effective protection for children and families experiencing severe poverty, in order to prevent long-term harm (Zamora‐Sarabia et al., 2019).

Materials and Methods

Study Design and Population

There is a cited work that describes the sample selection, although it is worth emphasizing that this article only includes the first phase of the study (without follow-up). The size of the study group also differs (1138 participants vs 836 participants). Please complete the text and provide appropriate explanations.

Thank you for your comment. The minimum necessary sample of 1138 is for the main outcome of the study (prevalence of overweight/obesity). This is a secondary study of that database. Following your comment, we have added more information on the missing information for each variable as follows: “Of the initial 1378 adolescents (100.0%) from the EDHLA study, 460 (33.4%) were removed from the study due to missing information about food insecurity. Additionally, 36 participants (2.6%) were eliminated due to a lack of data regarding sleep-related problems. Furthermore, 46 participants (3.3%) were excluded because of missing information on any covariate (i.e., body mass index, sedentary behavior, or energy intake)”.

Procedures

Lines 105-107. In my opinion, creating a category of “food insecurity” (category “very low food security”, and “low food security” together) is not a good solution from the perspective of further analyses because it may make it difficult to see the differences between categories: food security vs food insecurity. It would be better to look for differences in sleep problems in the group of 1/ food security vs. very low food security and 2/ food security vs. low food security. It seems that further analyses could be more interesting if the variable food security was treated as a continuous variable.

Thank you for your comment. This categorization has been used in other studies in Spain (Barreiro-Álvarez et al., 2024; Shankar-Krishnan et al., 2021). Since both categories (“very low food security” and “low food security”) are “undesirable” categories we decided to combine them. Since the ideal is to have food security (and not the other two categories), we have decided to keep the analyses as they are. On the other hand, we have already presented the analyses according to the food insecurity scale score, which provides a better understanding of the results.

Results

Lines 145-148. Are these differences significant? Why is there no p-value presented in the table, which would allow for an easy assessment of the obtained results?

Thank you for your comment. No p-values are included because that is not the objective of the study. To include p-values is to yield a result that should be discussed in the paper and is not in line with the objectives. Many journals suggest (for quite some time now) not to do so (unless it is an objective of the paper itself). We show the segmented data so that the reader can perceive the existence or not of large differences between the adjustment variables. If the editor deems it appropriate, we can include them.

Line 158. It is necessary to specify what these covariates were.

They have been included. Thank you.

Line 158. A new paragraph should start here or Table 2 should be cited at the beginning because this information is presented there, and not in Figure 1.

Thank you for your indication. The figure shows the increase in predicted probability as a function of each CFSSM-S point. The table shows the exact (numerical) value with the confidence intervals and the p-value. However, we have placed at the beginning the sentence referring to Table 2. Also, we have modified the wording for better understanding.

Figure 1. The quality of this Figure should be improved, e.g. it is difficult to read the variable name descriptions.

We have sent the figures as attachments (in pdf) as well as embedded in the manuscript for better quality. In the final version they should be included in this way. Thank you.

Table 2. It may be worth rethinking the table layout. Is it necessary to repeat CFSSM-S (per one point) for each variable? I suggest adding CFSSM-S score to the CFSSM-S description.

Done. Thank you.

Figure 1 and Table 2. Please explain whether these results are not about the same thing. Is it necessary to show them in two different graphic forms? Wouldn't a table be sufficient? A similar comment applies to Figure 2 and Table 3. Why is statistical significance not shown in Figure 1 as in Figure 2?

Thank you for your comment. You are partly right. The results correspond to the same thing. However, the figures give more information about the predicted probabilities as a function of each point obtained on the scale (Figure 1). As for Figure 2, the results are in predicted probabilities (percentage) instead of odds ratio, which might make it easier for the reader to interpret. If the editor deems it appropriate, we can leave the tables as supplementary material.

Discussion

The discussion is largely in the form of a literature review, e.g. lines 257-279, because there is no direct connection between this information and the results of our research. The difficulty in discussing the results may be related to using a general indicator to assess food security. It may be worth presenting the components of the CFSSM-S in more detail in the Result section, and then comparisons with the results of other studies will be easier.

Thank you for your comment. We have included a new table (Table 3) indicating the association between each item of the CFSSM-S and each sleep-related problem. The next information has been added: “The GLMs (using a backward stepwise regression) examining the association between each specific item of the CFSSM-S and having each sleep-related problem (adjusted for several covariates) are found in Table 3. Feeling worried (about running out of food) and skipping meals were associated with sleep problems (p=0.027 and p=0.036, respectively). Feeling worried (about running out of food) and hungry were related to excessive daytime sleepiness (p=0.017 and p=0.013, respectively). The lack of balanced meals was linked with awakenings during the night (p<0.001). Eating smaller portions was related to regularity and duration of sleep (p=0.031). Lastly, the lack of balanced meals and feeling hungry were associated with having any-sleep related problem”.

Lines 287-288. The large group is a strength, but the selection of the group (small region) does not allow for generalization to the entire population of young Spaniards and this is a limitation of the study.

Thank you for your comment. We have included this limitation as follows: “However, the data pertain to a very specific region of Spain, so they are not generalizable to the rest of the Spanish population”.

Conclusions

I suggest formulating conclusions in more detail, i.e. to indicate what sleep problems are associated with food insecurity.

The next information has been added: “Specifically, adolescents with higher food insecurity were more likely to have bedtime problems, excessive daytime sleepiness, awakenings during the night, or any sleep-related problem”.

Round 2

Reviewer 2 Report

Comments and Suggestions for Authors

Thank you for using my comments to improve the manuscript. I accept the responses except for one, namely:

Results: Lines 145-148. Are these differences significant? Why is there no p-value presented in the table, which would allow for an easy assessment of the obtained results?

Thank you for your comment. No p-values are included because that is not the objective of the study. To include p-values is to yield a result that should be discussed in the paper and is not in line with the objectives. Many journals suggest (for quite some time now) not to do so (unless it is an objective of the paper itself). We show the segmented data so that the reader can perceive the existence or not of large differences between the adjustment variables. If the editor deems it appropriate, we can include them.

I would definitely expect such information, the reader should not judge the differences between variables without such information.

Author Response

Thank you for your comment. P-values have been added.